# Combined Methodologies for the Survey and Documentation of Historical Buildings: The Castle of Scalea (CS, Italy)

**Eugenio Donato** [1,†] **and Dario Giuffrida** [2,*,†] 

1   Dipartimento di Civiltà Antiche e Moderne, Università degli Studi di Messina, SS. Annunziata, 98168 Messina, Italy
2   Institute for Chemical and Physical processes, National Research Council (IPCF-CNR), Viale F. Stagno d'Alcontres 37, 98158 Messina, Italy
*   Correspondence: dario.giuffrida@ipcf.cnr.it
†   These authors contributed equally to this work.

**Abstract:** In the last few years, new technologies have become indispensable tools for specialists in the field of cultural heritage for the analysis, reconstruction and interpretation of data but also for promotion of artefacts or buildings sometimes inaccessible or in a bad state of conservation. The discipline of geomatics offer many opportunities and solutions for integrated digital surveys and the documentation of heritage (point-based methods, image-based photogrammetry and their combination): These data can be processed in order to derive metric information and share them using databases or GIS (geographic information system) tools. This paper is focused on the description of combined survey methodologies adopted for the geometric and architectural documentation of the site and surviving structures of the Castel of Scalea (Cosenza, Italy). It is a typical context where traditional survey procedures do not fully succeed or require a longer amount of time and great effort if a high level of accuracy is requested: For this reason, aerial close-range digital photogrammetry enhanced by the GNSS (global navigation satellite system), and total station positioning systems have been used at various levels of detail for the production of a detailed 3D model and 2D thematic maps with an excellent level of in the positioning of the structures and in the architectural drawing. Thanks to the collected dataset, it was possible to better identify the building units (CF), to digitize the limits of the masonry stratigraphic units (USM), and to draw up a first constructive diachronic sequence hypothesis on which to base chronology. Moreover, some particular masonry techniques have been sampled and compared at the regional level with the aim to better dating of constructive expedients. It was finally demonstrated how the use of integrated methodologies allows us to obtain a complete and detailed documentation including information regarding not only architectural and geometrical features but also archaeological and historical elements, building materials and decay evidences—all useful as support of the interpretation of data.

**Keywords:** archaeology of architecture; Encastellation; digital photogrammetry; survey; 3D model; geometric documentation; diagnostic study; thematic maps; cultural heritage

## 1. Introduction

The current study is part a wider research project concerning the Encastellation phenomenon and the settlement on Mediterranean feudal areas, started in the end of the 90s by the Chair of Medieval Archaeology of the University of Florence [1] and continuing to date with several papers by E. Donato [2–4]: One of the topics of this research was the retracing the steps of the Encastellation process of the Tyrrhenian coastal area of Calabria between the Norman period and XIX century. In the

sub-regional zone between Capo Scalea (northern border) and the Savuto river (southern border), these studies have identified 14 castles (Figure 1a), one for the main centre and several isolated towers located along the stages of a north–south coastal route dating back to the Trajan age (in particular to the junction with the transverse route towards the hinterland) or in proximity of harbours. It is, however, plausible that the system also included other important fortresses mentioned by the sources (such as the Castrum Fellae, Cetraro and Fuscaldo), whose remains have not been still identified on field. In many cases the available documentation allowed us to follow the salient phases of the settlement history of this fortress, starting from the conquest of the Tyrrhenian Calabria by Normans until the defeat of the last Bourbon resistances against the French, in 1806, when all the castles, quashed their function, suffered definitive abandonment and spoliation processes.

Our research fits into this chronological range and aims to add a piece to a complex historical mosaic, such as the Encastellation of Southern Italy, which represents not only a solution for the organization and management of the territory but also for the protection and control of the viability.

The site of Scalea (Figure 1b,c), perhaps fortified before the arrival of the Normans [5] and conquered in 1057 by Roger de Hauteville as the first stage of the unification process of Southern Italy to the Kingdom of Sicily, is an integral part of the described defensive network: Its strategic function is linked to the presence, in ancient times, of an important seaport mentioned by Idrisi and illustrated on several portulan charts; it is also linked by the possibility to control a section of the road viability (via Capua-Regium) towards the actual Basilicata [6].

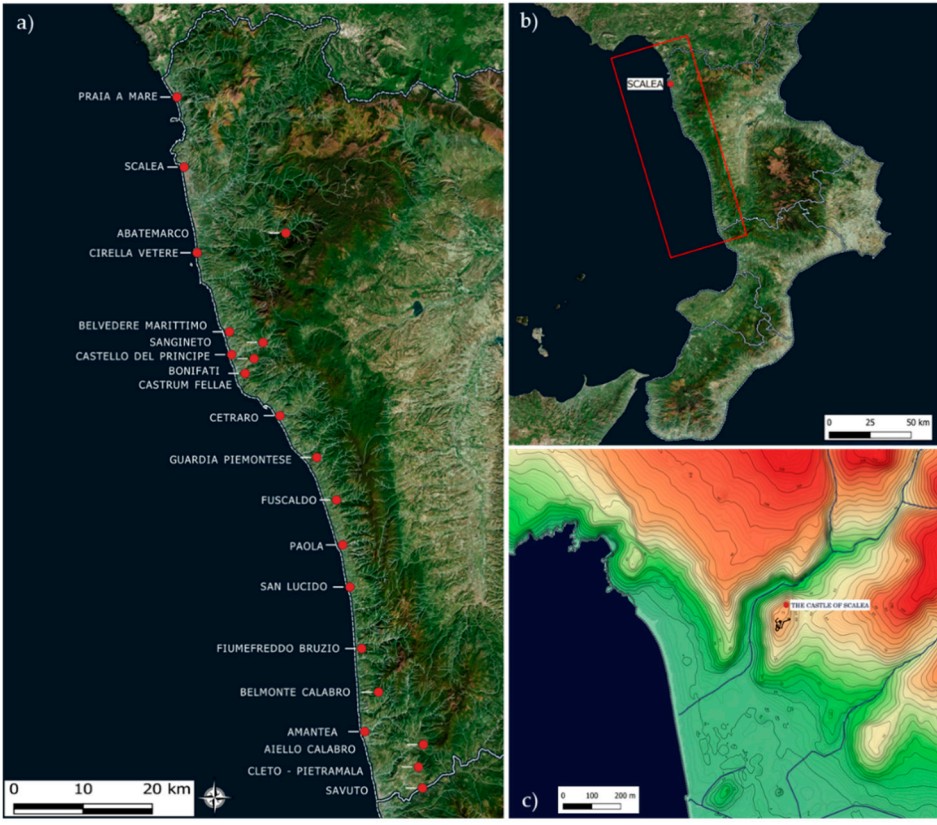

**Figure 1.** (**a**) The medieval defense system of the Tyrrhenian coast of Calabria between the Norman (XI century) and the Viceroyalty (XIX cetuty) (**b**) The Tyrrhenian coastal area of Cosenza. (**c**) Topographic map of the area of Scalea. (Images processed by authors).

It is necessary to premise that this paper, without any pretence of completeness, is configured as a synthesis of the methodological processes used for analysis and integration of data collected through different approaches on castles. In fact, only few data relating to some parts of buildings

have been reported by way of example, waiting for an overall and exhaustive edition of the historical, stratigraphic, and chronological data of which the two authors are already taking care.

*The Site and the Evidences*

The site rises on the top of a limestone dolomite slightly levelled artificially, which, with its elevation (80 m above sea level), dominates the village and the surrounding valleys (Figure 2).

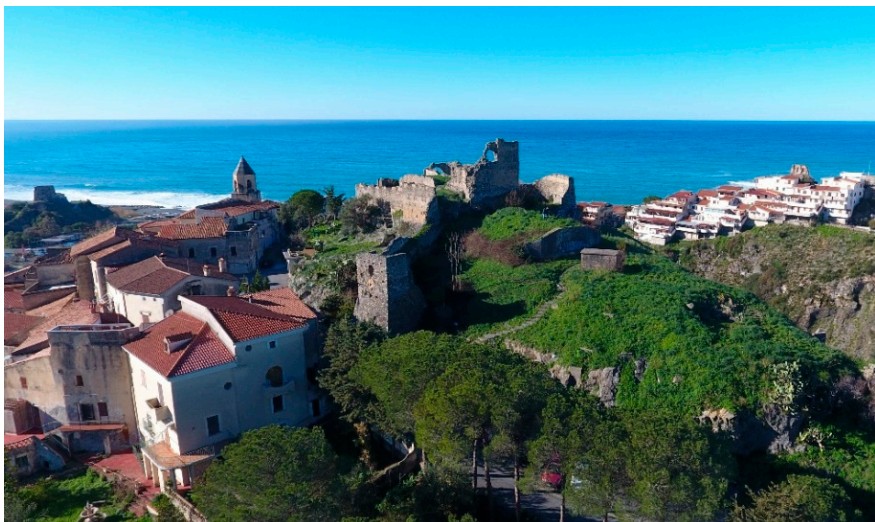

**Figure 2.** The ruins of the castle on the top of the hill (view from N-E).

The structures of the castle are characterized by an irregular plan, adapted to the morphology of the hill (Figure 3), with walls built upon the edges of the high ridges, especially along the western and southern sides.

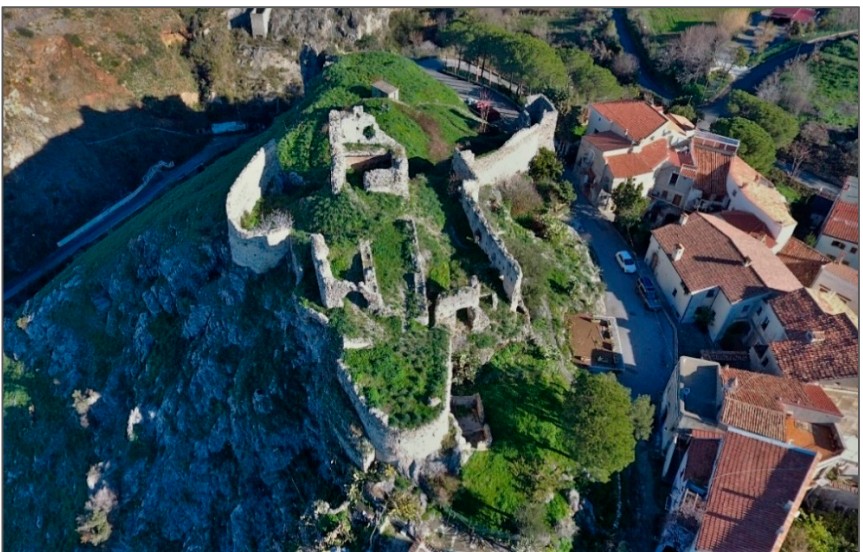

**Figure 3.** Aerial view of the hillfort from S-W.

Only a part of the original complex is preserved today: The currently visible perimeter of fortifications surround, in fact, exclusively the southern half of the hill (an area of about 1400 m$^2$), but it is conceivable that, in ancient times, the castle would extend to the north, following its natural boundaries.

The height of the preserved walls is imposing and, on elevation, it is possible to read that it is a complex stratification attesting the succession of the various construction phases (Figure 4). After the

Norman first installation, confirmed by historical sources, the structures were subjected to multiple interventions of a destructive and reconstructive nature which have sometimes altered the general layout of the complex up to the eighteenth century; what instead seems to have remained unchanged, judging by the historical archive documentation, is the strategic and defensive function, an element that sets it apart from all other complexes that, during post-medieval times, lost their defensive role.

Among its ruins, in 1908 (few years after the final abandonment), a large water basin was built connected to the first Scalea aqueduct. A legend tells that the castle is connected with an underground passage to the ancient Torre Talao, which stands on an imposing rock.

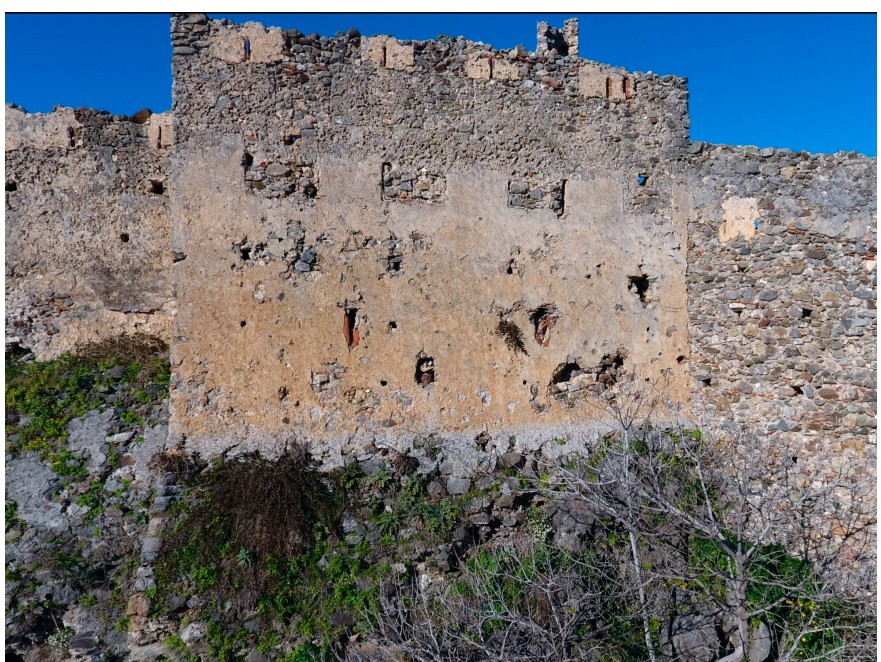

**Figure 4.** Example of a well-preserved multi-stratification on external elevation.

## 2. Materials and Methods

Different survey methodologies were adopted in order to respond to different study needs and according to various levels of detail (Table 1) [7].

**Table 1.** Survey methods.

| Type of Survey | Goals | Methods |
|---|---|---|
| Topographic | Creation of a base-map<br>Georeferencing<br>Positioning of structure | UAV<br>D-GPS |
| Architectural | Creation of elevation orthoimages<br>Drawing of single layer (USM) | Total station<br>Photogrammetry<br>Direct measurements |

An accurate analysis of the geo-morphology of the site, the evidences, the elevations stratigraphy and the construction techniques was carried out by integrating direct explorations to close-range photogrammetry and remote survey inspections, all with the aim to identify structures and to date, for the first time, the chronological phases of occupation and abandonment of the site.

The 3D architectural surveys were performed using two methodologies: Image-based and point-based modelling. The former, based on computer vision algorithms combined with photogrammetric procedures, was useful to obtain a photorealistic 3D model of the castle and to generate a new digital 2D base-map of

the site with contour lines; the UAV (**unmanned** aerial vehicle) system also helped in documentation of some structures (such as external elevation) not entirely accessible because of their locations. Differential GPS (with WGS-84/UTM32N coordinate system) was used to geo-reference the photogrammetric block and to precisely position the evidence. A total station (TST) was helpful to collect points for a smaller scale survey of diagnostic walls and to document architectural stratigraphic layers. Auxiliary manual sketches and eidotypes were also collected.

The achieved results were merged into a GIS (geographic information system) environment and processed for the purpose of obtaining thematic maps, while all the elevation drawings were performed in a CAD environment.

This approach, which could be defined as "global" [8] due to its interdisciplinary nature, in fact makes use of the historical-archaeological traditional tools of research (bibliographic research, study of written sources, stratigraphic analysis of elevations, periodization, study of historical buildings), adding the most innovative technologies useful for the surveying, management, and interpretation of data.

### 2.1. Site Surface Survey: Detection of Above-Ground Structures and Units of Building (CF)

A surface survey of the whole hillfort helped to gather information about all kind of activities on the site, identify and locate the above-ground structures and their features, and better plan the interventions. The analysis of the surviving structures of the castle was conducted through the following steps:

1. The detection, cataloguing and mapping of unit of buildings (Table 2).
2. Masonry stratigraphic analysis and the drawing of diagnostic elevations.
3. The construction of a chronological sequence starting from architectural stratification process of the castle from the 11th to the 15th.

In total, 11 building units (CF) were detected and mapped (Figure 5). Some of these were already known, while others were integrated through field survey and drone flights (e.g., CF 11).

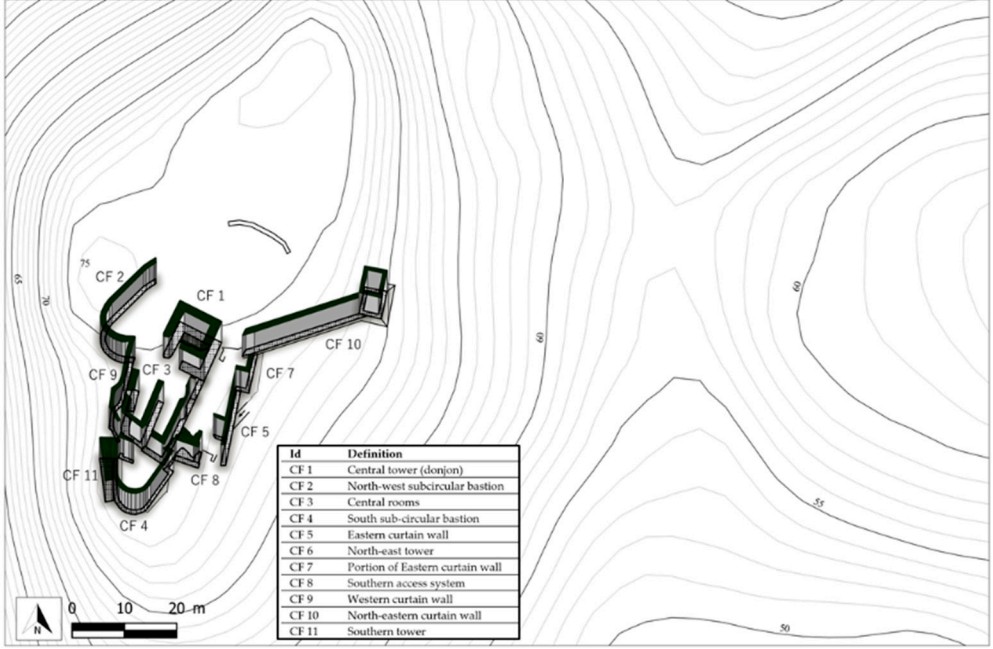

**Figure 5.** 2.5D view of the structure on topographic map (processed by authors).

**Table 2.** List of recognized topographic building units (CF).

| Id | Function | Description |
|----|----------|-------------|
| **CF 1** | Central tower (donjon) | Quadrangular keep (9.30 × 7.90 m) located at the highest point of the hill. The northern and western walls are still preserved for a maximum height of about 8.5 and 9 m, respectively. The tower was articulated on different levels: in the first level there was a cistern covered by a barrel vault, as shown by the layer of hydraulic plaster still visible in the walls; it is possible that, in the first phase, the first level housed a silo. There are two windows, one north in bricks and one in the east. On the south side, the tower binds to a wall with N–S course, which could be part of a first contemporary wall. |
| **CF 2** | North-west subcircular bastion | The perimeter masonry, visible for a length of 24 m, has a double row of embrasures along the internal elevation which were accessed by a wooden walkway (the holes for the supports in the masonry remain). |
| **CF 3** | Central rooms | Two adjacent rooms, with a N–S rectangular plan. The eastern room is accessible from E through a more recent scale system than the original walls, which are linked to the tower. The western room was accessible from the south, where there is a small embrasure in squared lava stone blocks. In the internal façade of the west wall, four putlog holes are visible, probably related to the support of a barrel vault roof and an opening: Its walls are slightly off axis compared to CF1. |
| **CF 4** | South sub-circular bastion | Sub-circular bastion, located on the southern edge of the castle, with the function of containing an artificial embankment. The drone survey along its inaccessible external facade highlighted the various constructive interventions not attributable, at the moment, to any chronological phase. Its construction incorporated CF11. |
| **CF 5** | Eastern curtain wall | Wall preserved for a length of about 15 m. It is characterized by a system of embrasures organized on two levels; the highest one is formed in the infill of the oldest battlements; outside the wall is protected by a talus added at a later date. |
| **CF 6** | North-east tower | Quadrangular tower characterized in the upper part by a first rectangular nucleus (4.30 x 5.30 m) to which a quadrangular talus was later added, bringing the overall dimensions to approximately 6.30 x 7.30 m. It had to be characterized by different levels. On the southern side, a machicolation is still visible, an element of defense of one of the accesses to the village that was located in that point. The tower has an articulated and complex constructive stratigraphy. |
| **CF 7** | Portion of Eastern curtain wall | Building equipped with talus, which continues the path of the CF5 wall towards N-N/E, but compared to this, it is more advanced than ca. 3 m. It is characterized by an interesting sequence of construction phases, but the presence of plaster does not allow for the stratigraphic analysis of the entire façade. The reconstruction of the battlements, which took place three times, is visible in the external perspective. |
| **CF 8** | Southern access system | System of access, probably related to the last phases of life of the complex, characterized by the presence of a small arched opening, set directly on the rock bench. |
| **CF 9** | Western curtain wall | North–south curtain, with irregular path. It follows the morphology of the plateau, connecting the western and southern bastions. At the level of the ground, only the crests are visible, while the view of its external facade is visible only from the outside and has been analysed by drone flight. To the south, there are the remains of an arch whose function is not at this time well defined. |
| **CF 10** | North-eastern curtain wall | Curtain wall (approximately 22 m long), preserved for a maximum height of 9 m. In the upper part, rectangular embrasures are visible. Covered at times by large patches of plaster (USM 5), the general external facade, on which the operations of survey and stratigraphic reading were concentrated, is characterised by a complex stratification that testifies to the numerous building phases that have alternated over the centuries. Three construction macro-phases have been identified. |
| **CF 11** | Southern tower | Building unit protruding about 1.90 m from the outer limit of CF 4 bastion; along the south-east facade there is a trace of a square curtain tower, relating to an earlier phase, which was probably incorporated by the construction of the bastion. |

## 2.2. Topographic Survey of the Hillfort

As a preliminary task for the study of the castle, a topographic survey was carried on the whole hill (an area of about 6000 square metres), with the aim of obtaining a geo-referenced and metrically correct digital model to be used as a working base and from which to generate DEM (Digital Elevation Model), contour lines, sections, and 2D orthophotos. For this step, close-range aero-photogrammetric acquisition technologies were combined with GNSS (global navigation satellite system) positioning systems according to an approach already successfully tested by authors during other works [9,10]. Respectively, we used a Phantom DJI 4 pro drone (equipped with a 12 mega pixel camera sensor, FOV (field of view) 94° 20–35 mm format equivalent) and a Topcon GR-3 GNSS receiver, equipped with a base and a mobile rover antenna set in the maximum configuration for the reception of GPS and GLONASS (GLObal NAvigation Satellite System) satellites [11].

A first flight was made keeping the camera in nadiral position at a height of 35 m and taking care to keep an overlap and sidelap between the frames of 75%. Maintaining this distance from the area, the area was covered with 130 images (Figure 6). Finally, a network of ground control points (GCP) was planned and measured by differential GPS (in *Real Time Kinematic* mode) for the correction and the referencing of acquisition.

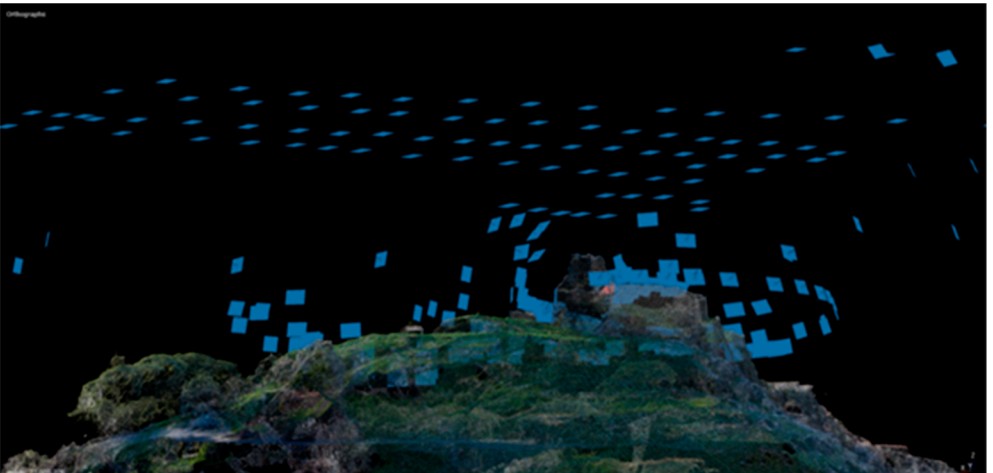

**Figure 6.** In blue the positions of cameras during the flight.

## 2.3. Architectural Survey: Wall Elevations

The structures have been documented combining oblique and zenithal images manually acquired according to a non-stereoscopic scheme [12], approximately 5 m off the evidence.

This method made it possible to obtain, in a short time (if compared to other techniques), a complete and detailed photographic coverage (because it was performed with low-altitude flights) of all the structures and to visually reach parts of the buildings that, due to their positions, were partially inaccessible (e.g., CF11, the external elevations of CF2 and CF4 towers, and the CF9 wall).

For each diagnostic elevation, measurements via a laser total station (Mod. Topcon GPT 1004) were also performed in order to acquire the necessary data for the creation of drawing based on ortho-photo, on which to represent the results of stratigraphic analysis (Figures 7–9).

Any other information regarding masonry techniques and materials was finally noted in specific tabs and sketches: During the stratigraphic analysis phases, a sampling of masonry techniques and a mapping of the distribution of bricks was also performed (Figure 7).

These vast array kinds of data were elaborated and integrated into the architectural drawings, developing thematic maps that recorded and represented the current preservation state of the monument and concerning major construction phases, building materials and decay.

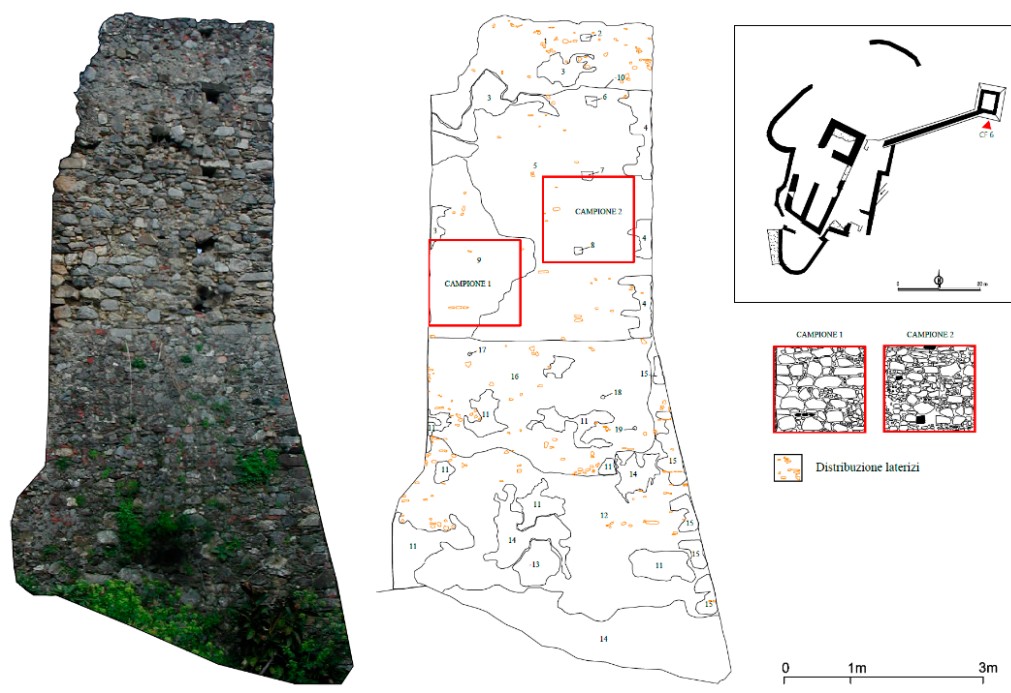

**Figure 7.** Thematic map: Distribution of bricks and plaster of CF6 elevation. The numbers indicate the architectural stratigraphic layer (USM).

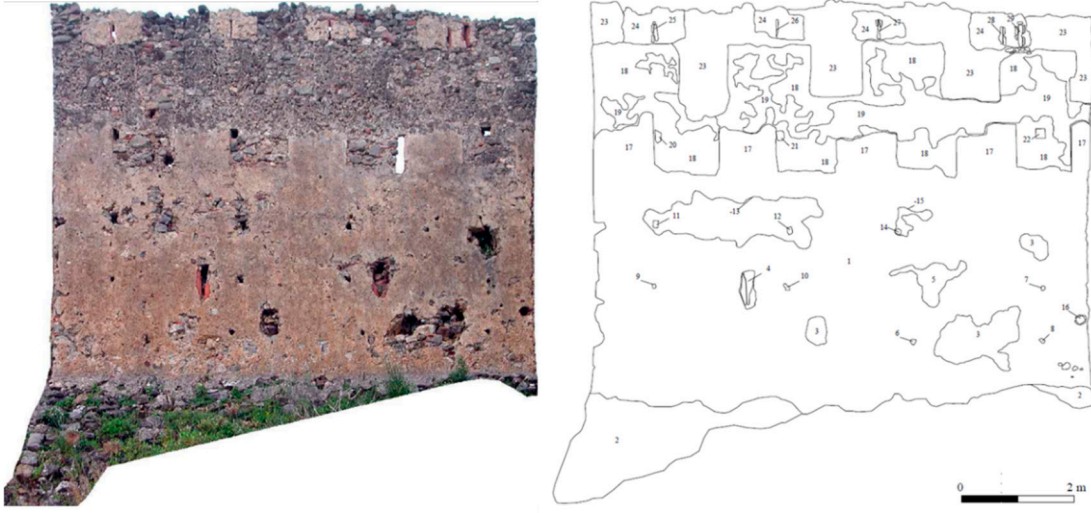

**Figure 8.** CF7: Orthoimage-based drawing and stratigraphic analysis.

Some Examples: The Documentation of CF1

On CF1, the stratigraphic analysis was focused on the best preserved western and northern elevations, from which an interesting stratification emerged (see Figures 9 and 10): It is interesting to observe the edge between the two walls (USM 1) characterized by a masonry in well squared blocks of lava stone and calcarenite set in parallel courses and laid in alternate rows of stretchers and headers. These corner blocks seem to be in phase with curtain wall portions (USM 10) consisting of a masonry made of split or rough-hewn stones, installed on courses that tend to be horizontal and parallel, where the horizontal courses are placed at intervals of about 40 cm. Numerous wedges consisting of stone chips and rare bricks can be seen at the joints of the blocks. This type of masonry, made by volcanic stone blocks, can be found also in the S/E angle and in the N/E one.

The other type of masonry visible on USM 11 seems to belong to a different phase: It is characterized by split stones installed on courses that tend to be horizontal and parallel; the horizontal courses are in

this case placed at intervals of about 35 cm and marked by rows of bricks and chips of stones, also present between the joints. This masonry is visible along the entire surface of the two elevations (USM 4). Another interesting feature is the presence of another irregular type of masonry (USM 3), recognised in the upper part of the elevations and perhaps attributable to a reconstruction of the upper part.

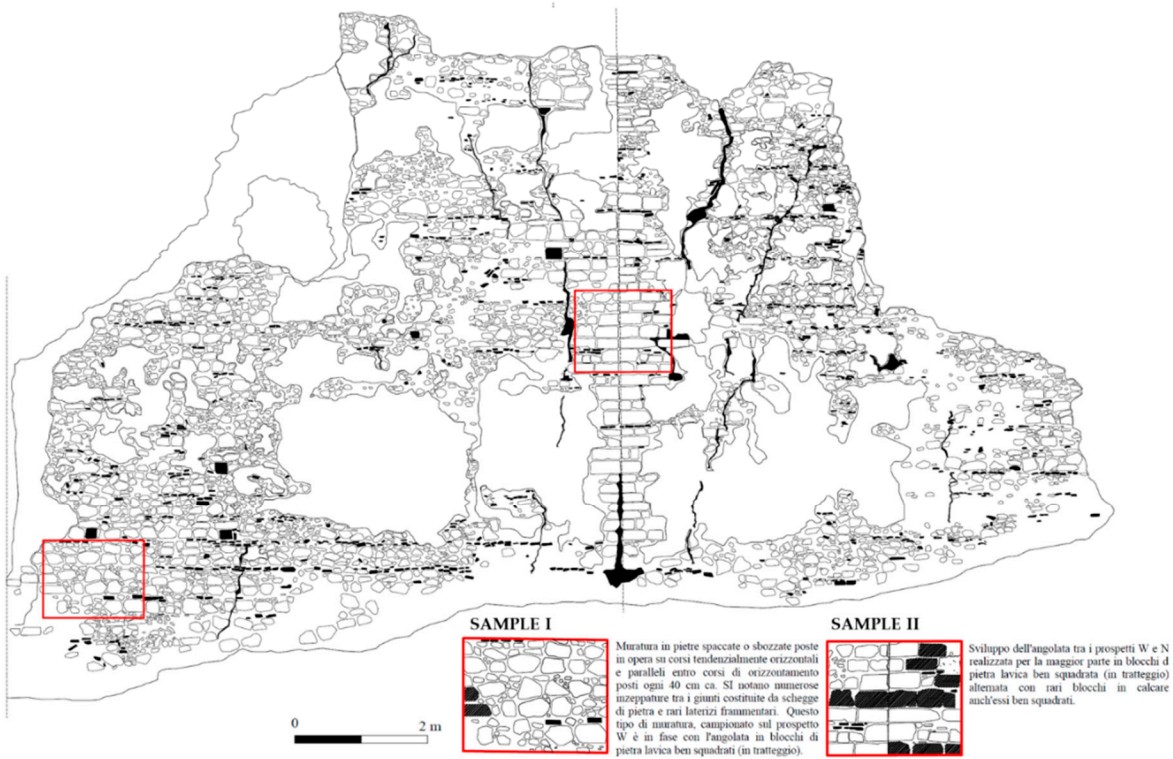

**Figure 9.** Western and northern elevation of CF1: Example of a CAD (Computer-Aided Drafting) drawing and sampling of masonry techniques.

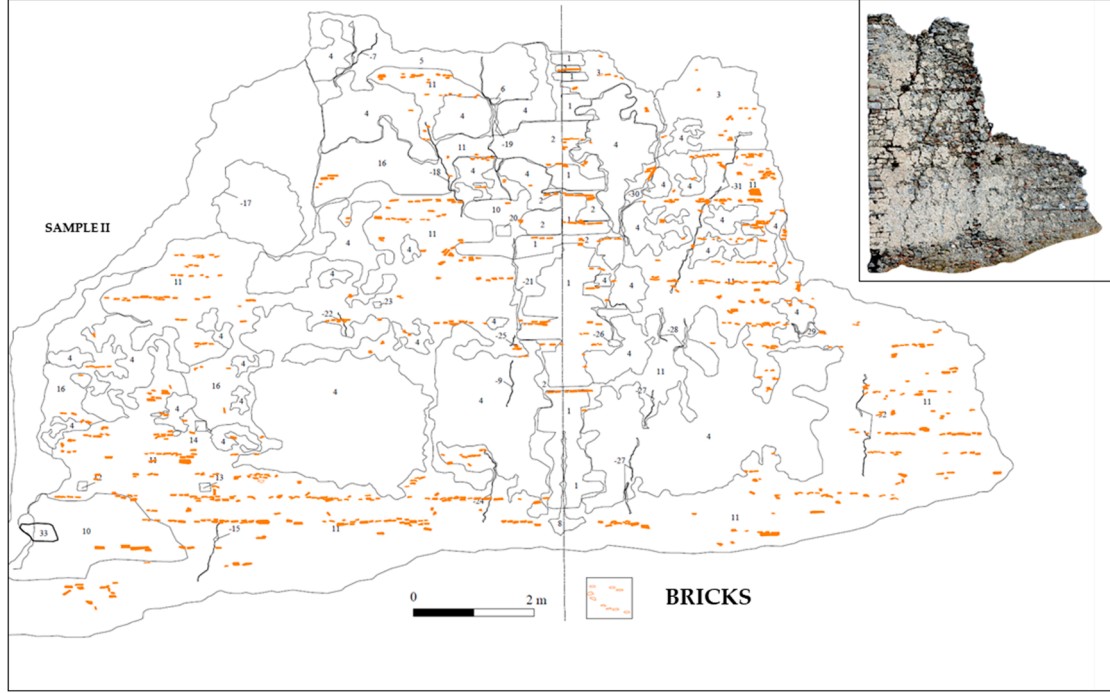

**Figure 10.** Thematic map: Distribution of bricks and plaster of CF1 elevation. The numbers indicate the USM.

### 2.4. Data Processing and Post-Processing

One of the advantages of the image-based survey is linked to the possibility of exploiting the algorithms of structure from the motion *(SfM)* range imaging technique, which is able to reconstruct the surface and volume of a given object in the three dimensions through the automatic collimation of common points between adjacent frames. The following processing steps were performed in Agisoft® Photoscan software [13]:

(1) Preliminary calibration of the camera using Lens, an Agisoft® plug-in [14], in order to reduce the error generated by the optical distortion during processing.

(2) Alignment of camera and creation of tie points (sparse cloud) between the frames: High quality and pair pre-selection was set; the process was based on reference (i.e., the position given by drone GPS).

(3) Creation of dense cloud was (Figure 11a) (setting medium quality and deep filtering on moderate).

(4) Creation of the three-dimensional model (mesh) starting from the point cloud (Figure 11b).

(5) The generation of the texture enabling adaptive orthophoto parameter: This parameter caused the horizontal and vertical surfaces to be treated separately (the first were texturized by orthographic projection, while the latter, like the elevation, were textured starting from oblique images.

(6) The semi-automatic classification of dense cloud points in order to exclude the elevations given by non-ground points from the calculation of the digital elevation model (DTM).

(7) Subsequently, a DTM was produced.

(8) Vector contour lines were generated with 1 m of equidistance and exported to a shp file.

(9) A complete GEOtiff orthophoto (reference system WGS-83/UTM 33N) was finally generated, which was useful for updating the planimetry of the structures.

Finally, a new up-to-date building plan (see Figures 4 and 12) was digitized on the new cartographic base-map.

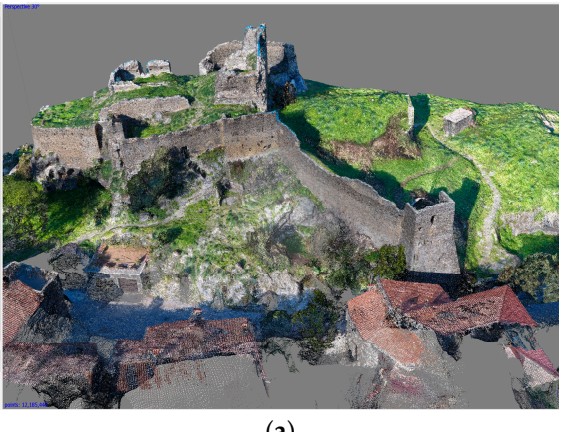 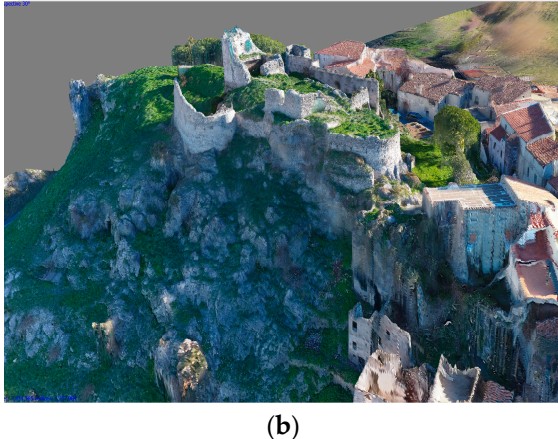

| (**a**) | (**b**) |

**Figure 11.** (**a**) Dense cloud model: View from the east. (**b**) 3D textured mesh model: View from the south-west.

### 2.5. Database and GIS Management

For each building unit, a digital data sheet was generated to record all the alphanumeric and spatial and information using MySQL®, a free and open source relational database management system, suitable for desktop and web-based applications. The database was connected to Qgis (GPL software, version 3.6.3) and implemented with architectural and building materials data, geometric documentation (vector drawings) and other types of information with the aim to create a dynamic 'integrated archive.'

## 3. Results and Discussion

### 3.1. Documentation Data

The deployment of combined close-range aero-photogrammetry, combined to traditional point-based methods (total station and Differential GPS positioning technologies) proved to be essential to get, in a short time, all the necessary tools for supporting our analysis and data integration [15,16]. In particular:

(1)  A high-resolution photographic coverage of the whole site (made of overlapping stereoscopic pairs) usable as a tool for monitoring the state of conservation.

(2)  A basic cartography of the geo-referred and ortho-rectified site to be used as the basis of a GIS for the study of the whole area: This is available in 2D versions (as a plan in GeoTIFF format) or in a scalable 3D model in suitable formats to be included in other analyses, developments and creation platforms for the promotion of multimedia products.

(3)  A database of scalable digital (CAD) drawings of the diagnostic elevations, useful both as a basis for our stratigraphic survey and for any digital reconstruction/restoration.

(4)  A detailed 3D model (dense cloud and triangulated irregular network ) for metric measurements and for the realization of multimedia products (education and virtual visit).

The advantage of the drone survey lies not only in the ever-increasing economic affordability of high-performing equipment and in the semi-automated *sfm* processing but, above all, in the quality of the photorealistic restitution, comparable metrically and geometrically to those obtained by scanning systems methods [17] that are much more expensive and complex in the data processing phase.

In relation to our objectives, we can conclude that these methods have proven to be appropriate for a better understanding of the castle, having allowed us to update the plan of the complex (Figure 5) to grasp some unknown construction problems and confirm others already known: From the geo-morphological analysis of the site (as we can see on the DTM, Figure 12), it is possible, for example, to assume the presence of structures even in the northern half of the hill, which is free of evidence today.

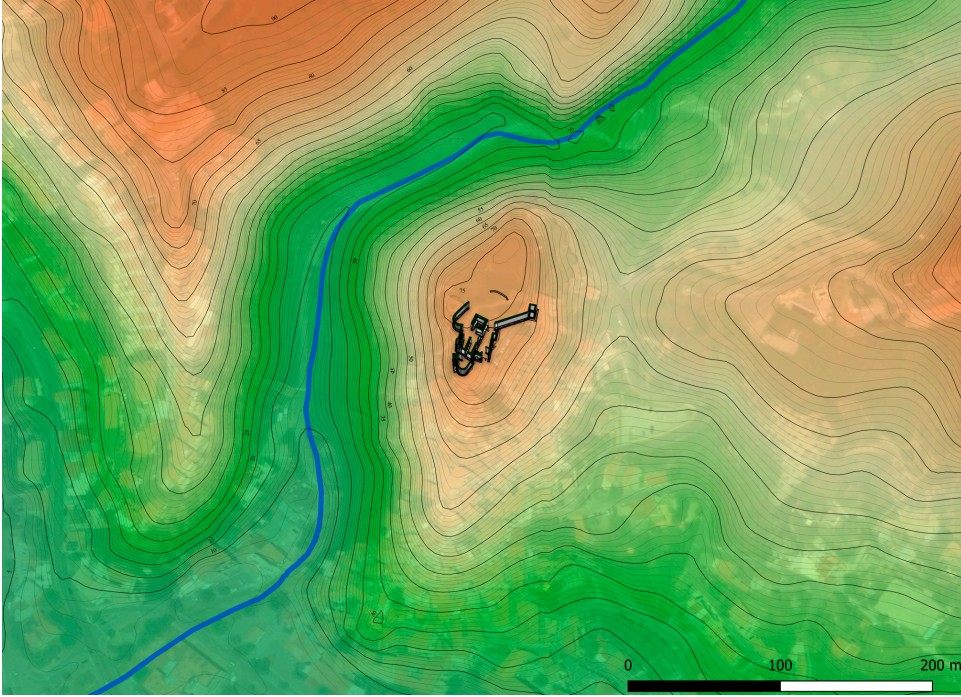

**Figure 12.** 2.5D rendering of structure on the digital elevation model (DTM), contour line base-map, and Google satellite images.

*3.2. Archaeological Data*

The study did not succeed in the detection of pre-Norman/Byzantine phases, highlighted in other fortified centres such as San Michele sull'Abatemarco, Pietramala, Amantea, Savuto and Aiello.

It is not clear if it is to be interpreted as a document of a Norman foundation of the site, to a lack of research, or to the failure to preserve the structures: Only future and more in-depth investigations (excavation or geo-physical prospection) will be able to verify the presence of previous phases.

In the regard of the preserved structures, we must point out the extreme situation of abandonment and degradation of the complex, which has not always allowed us to safely carry out works. The analysis of the construction sequence of the buildings based on the stratigraphic relationships allowed us to elaborate a first unpublished chronology of the main phases of the complex based on a regional comparison of the construction techniques.

In total, four main construction phases were distinguished (Figure 13): Even if the stratification is much more complex and suggests a greater number of phases, only certain data have been reported in the current conditions, postponing further distinctions to future insights.

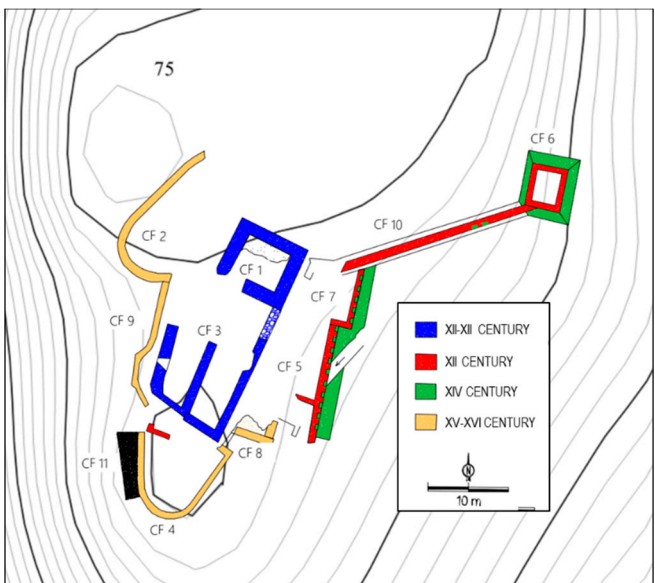

**Figure 13.** Schematic chronological map.

The oldest recognized building nucleus belongs to the Norman quadrangular tower (donjon) built in direct contact with the natural rocky bank, which is linked to the remains of a first curtain wall that surrounded the higher plateau: It is possible to attribute its construction to the earliest stages of the conquest by Roger de Hauteville, who, in accordance with the sources, chose to use the site as a base for raids in the territory. With the exception of the imported volcanic stone inserts, the stone material was extracted from the calcareous conformation on which the site stands.

In the Swabian phase (early XIII century) the castle was enlarged according to new planimetric models and construction techniques: Walls were built with battlements (CF 7), fortified by quadrangular towers (only the CF 6 tower has been preserved).

The model introduced is the "castle-fortress," characterized by a quadrangular plan and projecting towers whose origins date back to the type of the Roman castrum or to examples of eastern military architecture.

Between the XIII and XIV centuries (Angevin period), works were carried out to restructure and reinforce the structures: The raising of the curtains (on the elevation of CF 7, the Swabian phase was followed by a raising of the curtain with reconstruction of the battlements), the reconstruction of the battlements, and the addition of talus (CF 6) have been recognized.

To a fourth post-medieval macro-phase (VXI century) belong the two large sub-circular bastions on the east and north sides (CF 2 and 4), which are characterized by a more irregular masonry,

Taking into account the data coming from the most recent research on the region, we can conclude that the settlement-architectural developments of the Scalea castle fit fully into the general framework: Building typologies and settlement dynamics were widely compared at the regional and extra-regional levels (Basilicata). However, it is necessary to specify that the general data of the sites embedded in Calabria are anything but definitive.

## 4. Conclusions and Future Developments

The multidisciplinary methodological process described in this paper has proven to be suitable to achieve the initial goals: The 3D and 2D geometric and graphic documentation provided information regarding building techniques and materials used in the exterior façades and helped in identifying all the construction phases of the castle. This kind of combined documentation, based both on traditional and innovative instrumental surveys, represents (as well-demonstrated) a valid tool supporting the interpretation and analysis of historical buildings.

On the other hand, even if the precision on the measurements obtained is not enough for the architectural monitoring of structures (this was not our goal), it is enough for creating an exhaustive, representative 3D model of the monumental complex which is useful for the fruition and diffusion of the knowledge of heritage. For this purpose, the active contribution of these technologies is clear, since the graphic documentation is an essential moment of investigation.

In conclusion, all these products obtained represent a fundamental tool of specialised knowledge in the archaeological field, supporting stratigraphic, monitoring and diagnosis for conservation or future restoration activities. Nevertheless, they are already saved in a format suitable for embedding in GIS and H-BIM (Heritage Building Information Modeling) environments, where they can be managed and implemented with further research.

**Author Contributions:** Conceptualization, E.D.; data curation, D.G.; investigation, D.G.; methodology, E.D.; project administration, E.D.; supervision, E.D.; visualization, D.G.; writing—original draft, D.G.; writing—review & editing, E.D. and D.G.

**Funding:** This research received no external funding.

**Conflicts of Interest:** The authors declare no conflict of interest.

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
