# Peer review of "Combined Methodologies for the Survey and Documentation of Historical Buildings: The Castle of Scalea (CS, Italy)"

_heritage, doi:10.3390/heritage2030146_

Round 1
Reviewer 1 Report
It is a nice piece of work. Well written and structured. I would suggest considering two significant points to consider to enhance readership -
1. The paper claims, "The methodological process is the main outcome of this work .......". The point is this claim needs a substantial background study of the current works with pros and cons, and an explanation of why the proposed methodology is unique from others. Section 2, therefore, needs significant revision.
2. Because this paper aims to present and highlight a unique methodology. Readers might not be interested in what has been found in a specific site (section-3), rather critical analysis of the used methods. Explain their shortcomings and strengths in surveying the heritage site and how these have been overcome in this project. This will benefit the readers to use the proposed/presented methodology on other heritage sites.
Few minor issues-
1. Please use the abbreviation of all acronyms. Readers from other domains may not know what CF etc. is.
2. In general photos, figures and tables are sourced from/by the author. So, no need to mention 'photo by authors', which is not borrowed from others.
3. A diagram/flowchart would be beneficial for the readers at section-2, to understand how each method follow other methods to complete the methodology; and the flow of the information in various step.
Author Response
Dear reviewer,
Thanks for your revision.
With reference to the comment n. 1, we changed this sentence from "The methodological process is the main outcome of this work .......", to “The methodological process adopted for this work has proved to be suitable to meet the initial objectives […]”
With reference to the comment n. 2, we believe that the reader is instead interested in also obtaining archaeological information about the site object of the survey: the methodology that was adopted is not in fact merely aimed at producing 3d models and 2d plants, but has contributed, together with the stratigraphic analysis, to better understand, classify and document the structures (specially for the less accessible parts).
There are many articles that are used to evaluate the pros and cons of these relevant methodologies, some of which have been reported in the bibliography. However this was not the fundamental objective of the present work.
With reference to the “minor issues”:
1. All the acronyms in the text have been better specified and clarified.
2. The sentences ‘'photo by authors'’ below the images has been now deleted.
3. Even if no flowchart was inserted as an image, the workflow of each process was detailed in the paragraph dedicated to data processing.
Reviewer 2 Report
it is an applied paper on issues of cultural heritage. There is a lot of work done in this area. In my opinion the authors have to emphasize what is new in this paper. A big disadvantage is that the references are only Italian. You must add international references.
Some recommended references are:
1. K. Bacharidis, F. Sarri, V. Paravolidakis, L. Ragia, M. E. Zervakis, 2018 : Fusing Georeferenced and Stereoscopic Image Data for 3D Building Facade Reconstruction: ISPRS International Journal of Geo-Information 7(4): pp. 151-173. DOI: 10.3390/ijgi7040151
2. Three-Dimensional Digital Documentation of Cultural Heritage Site Based on the Convergence of Terrestrial Laser Scanning and Unmanned Aerial Vehicle Photogrammetry ISPRS Int. J. Geo-Inf. 2019, 8(2), 53; https://doi.org/10.3390/ijgi8020053
3. To 3D or Not 3D: Choosing a Photogrammetry Workflow for Cultural Heritage Groups
Heritage 2019, 2(3), 1835-1851; https://doi.org/10.3390/heritage2030112
other remarks are:
line 41 [1.] delete the point
line 56 for two times
line 67 cent and cec
line 118 global I would not mention that. Global as meaning is difficult to use it in such an approach.
line 128 () empty brackets
line 132 CF you much define before what it means
line 132 the references to figures must be unified. Sometimes they are figure and sometimes fig.
line 141 square meter must be written m2
line 145 references [9] [10] are in bold
line 216 points1 what does it mean?
line 235 there is close range photogrammetry and aerial photogrammetry no other definition
line 252 correct the engish grammar these methods have proved not has ..
line 304 correct the englich grammar precision is not are
You have to control the english language and the editing mistakes
Author Response
Dear reviewer,
Thanks for your revision.
With reference to the comment “the authors have to emphasize what is new in this paper”, in the new text we have tried to better emphasize the new elements of the work, which are not given by the product of the survey , i.e. the 3d models (whose methodologies of execution are now a fact of modern archaeological documentation) but by the results in terms of chronology and history of the castle, to which new technologies have contributed together with traditional approaches.
However the following recommended references has been added:
1. K. Bacharidis, F. Sarri, V. Paravolidakis, L. Ragia, M. E. Zervakis, 2018 : Fusing Georeferenced and Stereoscopic Image Data for 3D Building Facade Reconstruction: ISPRS International Journal of Geo-Information 7(4): pp. 151-173. DOI: 10.3390/ijgi7040151
2. Three-Dimensional Digital Documentation of Cultural Heritage Site Based on the Convergence of Terrestrial Laser Scanning and Unmanned Aerial Vehicle Photogrammetry ISPRS Int. J. Geo-Inf. 2019, 8(2), 53; https://doi.org/10.3390/ijgi8020053
3. To 3D or Not 3D: Choosing a Photogrammetry Workflow for Cultural Heritage Groups
Even if “there is a lot of work done” in the field of 3d survey of historical buildings, the historical and chronological results obtained with the present research make the work unique and important in the context of the phenomenon of the Encastellation of Tyrrhenian Calabria.
Also the corrections in your list have all been made and English has been revised.
Best regards.
Reviewer 3 Report
Form
The title numbering needs to be corrected (1.2 without a mention of a 1.1, double mention of the same number e.a. 2.3).
Some sentences are incomplete and/or punctuation is missing.
Figures and tables are not in citation order.
Some citations are incorrect : not referring to the correct figure or citing a nonexistant figure (cf. fig. 13).
Text in most of the figures is too small to be read and understood.
Figure 6 is never cited and therefore has no use.
Table 2 is cited once, however text is referring to table 1. Therefore, table 2 is actually never cited and has no use.
Some maps are missing a scale and north symbol.
If abbreviations are used, they should be explained once next to the terms they are referring to in order to be easily understood by the reader.
The references are somewhat confusing, following the publication guidelines would make them easier to understand.
Content
Some of the words used seem to be misunderstood and used which lead to a lot of questioning while reading: "methodology" when "method" seems more appropriate, "plant" instead of "plan", "diagnostic wall / elevation", "sampling of techniques", ...
GIS and Photogrammetry are not really innovative technologies since they were developed a few decades ago.
The terminology "topographic unit of building" does not make sense.
Visually interpreting a DTM is a step that can be part of a geomorphological analysis , but it is not a geomorphological analysis in itself.
It is not clear if the photographic acquisition of the castle was done only via drone or not. IF the acquisition was only done by drone, where and how were the zenithal images taken ?
Is the example CF 1 considering north and west elevations (according to the text) or north and east elevations (according to the captions below the figures) ?
The example CF1 is quite complicated to grasp as the texts and numbers on the figures are unreadable.
Structure from motion is a technique/process which can make use of different algorithms, but it is not an algorithm in itself.
"aerial close-range aero-photogrammetry" seems quite redundant.
It seems that some technical terms are not fully understood and this causes confusion while reading the paper.
The authors mention a "global" approach, however the data essentially mentioned and exploited in this paper are the acquired images and the 3d models. A "traditional historical-archaeological research" and a GIS database are mentioned, but no further details are provided.
The text jumps from explaining the acquisition and processing of the images to the discussion of the results. Since the analysis of the elevations is not clearly provided, the reader cannot be critical towards the results as he does not have access to the data and drawings.
The discussion of the results does not refer to references/sources or provide any comparison between the information gathered the traditional way and the newly acquired data, which could also highlight the importance of adding other tools to the so called traditional ones. Also, no figure displaying an example of a preliminary stratigraphy is included which could help the reader to understand the succession of the construction phases.
The conclusions state that the "methodological process is the main outcome". However, the combination of "traditional historical-archaeological research tools" with technologies used in geomatics is a process that is more and more used for an archaeological investigation and therefore, it cannot really be considered as a novelty. But if the main goal was to put forward, display, explain and compare the newly gained data and information and thus providing new critical knowledge, there would be an interesting and original contribution.
The conclusions also mention that this kind of survey is not precise enough in order to monitor the structures. This part is addressed nowhere else in the paper and no clear explanation is provided for why the precision is not sufficient for such a purpose.
Author Response
Dear reviewer,
Thanks for your revision. We have tried to apply all the suggestions to the manuscript.
Form:
1. The title numbering has been corrected. 2.3.1 is a subparagraph of 2.3 (it is not the same);
2. The sentences have been completed and punctuation has been added;
3. Figure order has been fixed;
4. Text size does not depend on us;
5. Reference to figure 6 has been added to the line 172;
6. Reference to tables has been fixed;
7. Scale ad north has been added on figures;
8. The meaning of abbreviations has been clarified and explained;
9. The references have been adapted to the guidelines of Heritage.
Content:
1. The terminological question has been fixed;
2. Even if GIS and Photogrammetry are not innovative technologies, the application in contexts like this, aimed at studying the Encastellation of the Tyrrhenian Calabria, and the historical results obtained combining this methods with the traditional archaeological approaches is a novelty.
3. The terminology "topographic unit of building" has been replaced with building unit or unit of building
4. “If the acquisition was only done by drone, where and how were the zenithal images taken?” The structures have been documented through oblique and zenithal images manually acquired according to a non-stereoscopic scheme, approximately 5 m off the evidence (it is possible to see on image 6 also the position of cameras of zenithal images).
5. On the example CF1 considers western and northern elevations according to the text; the caption below the figure has been modified.
6. “The example CF1 is quite complicated to grasp as the texts and numbers on the figures are unreadable”: the numbers indicate the stratigraphic units recognized and which are at the base of the chronological scan of the buildings.
7. “Structure from motion is a technique/process which can make use of different algorithms, but it is not an algorithm in itself”: the sentence has been changed as following “ One of the advantages of the image-based survey is linked to the possibility of exploiting the algorithms of structure from motion (SfM) range imaging technique […].
8. "aerial close-range aero-photogrammetry" redundance has been fixed.
9. About a "global approach" in archaeology, a clarification has been provided to the note 8.
10. “The text jumps from explaining the acquisition and processing of the images to the discussion of the results. Since the analysis of the elevations is not clearly provided, the reader cannot be critical towards the results as he does not have access to the data and drawings.”
à The part relative to analysis of the elevations have been modified and an image was added.
11. Also, no figure displaying an example of a preliminary stratigraphy is included which could help the reader to understand the succession of the construction phases” à A complete stratigraphy is displayed on figure 13 as result of the work.
12. Conclusion has been revised.
13. The new elements of the work are not given by the product of the survey (whose methodologies of execution are now a fact of modern archaeological documentation), i.e. the 3d models or 2d plans, but by the results in terms of chronology and history of the castle, to which new technologies have contributed together with traditional approaches. Even if there are a lot of work done in the field of 3d survey of historical buildings, the historical and chronological results obtained with the present research make the work unique and important in the context of the phenomenon of the Encastellation of Tyrrhenian Calabria.
Best Regards
Round 2
Reviewer 2 Report
I agree with the corrections
Author Response
We have upload our correction. Thank you for your revision.
Reviewer 3 Report
Dear authors,
Thank you for clarifying some points and reacting to the comments.
However, the numerotation of the titles and subtitles is still off. The text inserted into the figures (for example, the numbers of the USM (figures 7, 8 and 10) or the samples explanations (figure 9)) are still too small to be read and understood. Figures 1b and 1c are still missing a scale.
Author Response
Dear reviewr,
Additional requested changes have been made: the numeration has been fixed; the scalebars on fig. 1b-c and 12 and have been added; the text size on samples explanations (figure 9) had been modified.
Please accept the publication
Best regards.